# HiReview: Hierarchical Taxonomy-Driven Automatic Literature Review Generation

## Abstract

In this work, we present HiReview, a novel framework for hierarchical taxonomy-driven automatic literature review generation. With the exponential growth of academic documents, manual literature reviews have become increasingly labor-intensive and time-consuming, while traditional summarization models struggle to generate comprehensive document reviews effectively. Large language models (LLMs), with their powerful text processing capabilities, offer a potential solution; however, research on incorporating LLMs for automatic document generation remains limited. To address key challenges in large-scale automatic literature review generation (LRG), we propose a two-stage taxonomy-then-generation approach that combines graph-based hierarchical clustering with retrieval-augmented LLMs. First, we retrieve the most relevant sub-community within the citation network, then generate a hierarchical taxonomy tree by clustering papers based on both textual content and citation relationships. In the second stage, an LLM generates coherent and contextually accurate summaries for clusters or topics at each hierarchical level, ensuring comprehensive coverage and logical organization of the literature. Extensive experiments demonstrate that HiReview significantly outperforms state-of-the-art methods, achieving superior hierarchical organization, content relevance, and factual accuracy in automatic literature review generation tasks.

## 1 Introduction

Literature reviews play a crucial role in synthesizing knowledge from large bodies of work, helping to organize and summarize relevant research. However, manual literature reviews are labor-intensive and time-consuming, especially when addressing fields with complex, hierarchical topics. Consequently, there is growing interest in automating literature review generation (LRG) to reduce this burden. As shown in Fig. 1, an automated LRG system should accurately retrieve relevant papers, identify relationships between them, organize the papers and generate reliable, factually accurate content for the literature review.

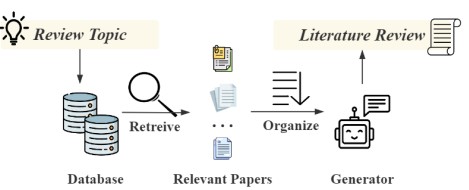

Figure 1: An illustration of automate literature review generation system.

Traditional multi-document summarization models (Abdel-Salam & Rafea, 2022; Erkan & Radev, 2004; Gunaratna et al., 2015; Izacard & Grave, 2020; Kasanishi et al., 2023) struggle to handle the extensive input lengths characteristic of scientific reviews, often resulting in shallow summaries that lack the depth required for comprehensive literature reviews. Recently, large language models (LLMs) have been applied to this task due to their ability to manage long-range dependencies and provide richer contextual understanding. However, directly employing LLMs in academic writing can lead to issues such as hallucinations—where the generated content is not grounded in factual data—and may fail to accurately reflect state-of-the-art research (Huang et al., 2023; Tonmoy et al., 2024). To mitigate these issues, some LLM-based approaches have integrated retrieval-augmented generation (RAG) to enhance content quality (Fan et al., 2024; Gao et al., 2023b). In LRG, tools like Paper Digest[1] can generate brief summaries but lack the depth and comprehensiveness required for detailed literature reviews. AutoSurvey (Wang et al., 2024) addresses the scalability challenges of

---

[1]https://www.paperdigest.org/review/

traditional review methods by employing a specialized LLM in a multi-stage process that includes retrieval, chapter drafting, and refinement. Similarly, Wu et al. introduced an LLM-based method for automated review generation, consisting of four key components: literature retrieval, outline formulation, knowledge extraction, and review composition. However, these LLM-based approaches overlook critical prior knowledge, such as citation relationships, which are essential for retrieving relevant papers and producing coherent, accurate literature reviews. Additionally, they fail to organize papers into meaningful structures before the generation phase, limiting the system's ability to fully comprehend the relationships between papers. These models rely too heavily on LLMs, with the generation process being primarily driven by prompts, resulting in less robust and often inconsistent outcomes.

To overcome the limitations of current LLM-based methods, we propose **Hi**erarchical Taxonomy-Driven Automatic Literature **Review** Generation (**HiReview**), which fully leverages prior knowledge from citation networks, such as the network topology formed by papers, rather than relying solely on text. To enable LLMs to better understand the underlying relationships between papers, we propose generating a hierarchical taxonomy of the papers before proceeding with content generation. We introduce an end-to-end framework that generates a taxonomy from a large citation network. Specifically, we explore the relationships between papers in a hierarchical manner, where papers are divided into clusters, and each cluster at every hierarchical level corresponds to a specific topic in a field. We then employ a hierarchical generation strategy to determine the central topic for each cluster, ensuring coherence across the hierarchy. By organizing papers into a clear, structured taxonomy, HiReview enables LLMs to better comprehend the relationships between papers, resulting in robust, high-quality, and citation-aware literature reviews.

In summary, our primary contributions are as follows:

- **Dataset.** We construct a large-scale dataset of literature review papers. Each paper is annotated with its hierarchical taxonomy and related citation network, which serves as the foundation for training and evaluating literature review generation models.

- **Framework.** We introduce HiReview, a novel hierarchical taxonomy-driven generation framework for literature review generation. Papers are first organized into a meaningful hierarchical taxonomy, which is then leveraged to guide the generation phase.

- **Robustness.** We demonstrate the robustness of HiReview across various experiments. Compared to baseline models, HiReview exhibits superior consistency and quality in literature review generation, including better coverage, structure, and relevance.

## 2 RELATED WORK

**Retrieval-Augmented Generation (RAG).** RAG has gained increasing attention for its ability to mitigate the hallucination problem in LLMs, thereby enhancing the trustworthiness of generated content (Chen et al., 2024; Gao et al., 2023b; Lewis et al., 2020). By dynamically retrieving relevant information from a large corpus during the generation process, RAG models improve the accuracy and relevance of outputs in applications such as citation-aware generation (Gao et al., 2023a; Menick et al., 2022), text evaluation (Xie et al., 2024; Yue et al., 2023), and open-domain question answering (Izacard & Grave, 2020; Karpukhin et al., 2020; Zhu et al., 2021). This approach helps ground the model's output in up-to-date and relevant information, reducing the risk of hallucination and improving the reliability of large-scale generative models. More recently, graph retrieval-augmented generation (Edge et al., 2024; Hu et al., 2024; Peng et al., 2024) has garnered widespread attention, with relationships between documents or chunks proving to effectively enhance the scalability and performance of RAG in graph-related tasks (He et al., 2024; Mavromatis & Karypis, 2024).

**Parameter-Efficient Fine-Tuning (PEFT).** Full-parameter fine-tuning of domain-specific large-scale pre-trained models requires significant resources. In contrast, parameter-efficient fine-tuning (PEFT) focuses on enhancing model performance in specific domains without updating all model parameters (Ding et al., 2023; Han et al., 2024). Key techniques in PEFT include Adapters, Low-Rank Adaptation (LoRA) (Hu et al., 2021), and Prompt Tuning. Adapters are small neural networks inserted into the layers of a pre-trained model to modify its behavior for specific tasks (Houlsby et al., 2019; Pfeiffer et al., 2020). Instead of updating the entire model, only the adapter parameters are updated, making the process computationally efficient. Adapters have also been explored in applying PEFT to graph-based language models (Chai et al., 2023; Liu et al., 2024; Perozzi et al.,

2024). LoRA introduces low-rank matrices into transformer layers, enabling fine-tuning of only a small subset of the model's parameters while preserving overall performance. Prompt Tuning involves learning task-specific prompts that guide the model to perform various tasks without altering its underlying parameters (Lester et al., 2021; Li & Liang, 2021; Liu et al., 2023). The pre-trained model remains fixed, with only the input prompts being optimized for the specific task.

## 3 PRELIMINARIES

The goal of this paper is to generate a structured, summarized review in response to a user's query. The process consists of three main stages: retrieving relevant papers, clustering papers based on shared characteristics, and generating a synthetic content summary. Before formally presenting the problem, we first define its key components to establish a shared understanding.

**Citation Graphs.** A citation graph is a directed graph where the nodes represent academic papers, and the edges represent the citation relationship between them. Formally, it can be defined as $G = (V, E, \{T_u\}_{u \in V})$, where $V$ denotes the set of nodes (i.e., scientific papers), $E \subseteq V \times V$ denotes the set of edges (i.e., citation relations). Specifically, $t_u \in T_u$ denotes the title of paper $u$. This representation enables the extraction of both structural and semantic information from the academic literature, forming the backbone of subsequent paper retrieval and clustering tasks.

**Retrieval-Augmented Generation.** To enhance the quality and relevance of the generated summary, we utilize a retrieval-augmented generation approach. Given a corpus of external knowledge sources $\mathcal{D} = \{d_i\}_{i=1}^{N}$ and a query $Q$, this method aims to leverage an optimal subset $\mathcal{D}' \subseteq \mathcal{D}$ to enrich the response generated by a PLM. Formally, the generative process is captured by the following probability distribution for the output sequence $Y$:

$$p_\Theta(Y|Q, \mathcal{D}) = \prod_{i=1}^{r} p_\theta(y_i|y_{<i}, X_Q, [X_d]_{d \in \mathcal{D}'}), \tag{1}$$

where $\Theta$ denotes the PLM's parameters, $Y$ is the token sequence in the generated response, $y_{<i}$ indicates the prefix tokens, $X_Q$ represents the token sequence of the query $Q$, and $[X_d]_{d \in \mathcal{D}'}$ denotes the concatenation of token sequences from the relevant external sources retrieved from $\mathcal{D}'$. This formulation ensures that the generated content is not only contextually accurate but also grounded in the retrieved academic literature.

**Hierarchical Graph Clustering.** A hierarchical graph $G = (V, E)$ is a multi-level representation in which nodes and edges are organized across various levels of granularity, leading to progressively abstract representations. Let $G_l = (V_l, E_l)$ denote the graph at level $l$ in the hierarchy, where $V_l$ and $E_l$ denote the set of nodes and edges at $l$-th level. The goal of hierarchical graph clustering (Ren et al., 2024; Xing et al., 2021; Ying et al., 2018) is to generate clusters $C_l = f(G_l)$ at each level $l$, using a clustering algorithm $f(\cdot)$. At each level $l$, the clusters $C_l = \{c_l^i\}_{i=1}^{|V_{l+1}|}$ are formed, where each cluster $c_l^i \subset V_l$ represents a subset of nodes. These clusters $C_l$ at level $l$ serve as hyper-nodes for the next level $l + 1$, forming a new graph $G_{l+1} = (V_{l+1}, E_{l+1})$, with $V_{l+1} = C_l$. The process is applied recursively, aggregating clusters at each level until a stopping criterion is met, i.e., no further meaningful clusters can be identified.

**Literature Review Generation (LRG).** Given the hierarchical graph clustering approach, we now present the formal problem definition of LRG. The task is to generate a comprehensive summary $\mathcal{R}$ based on a citation graph $G(V, E, \{T_u\}_{u \in V})$ and a research topic $Q$ as a query. The goal is to output $\mathcal{R} = \{R_k\}_{k=1}^{K}$, where each $R_k$ is represented as a sentence or paragraph that discusses the most relevant papers to the query. A high-quality literature review must exhibit *a clear structure*, often reflected through an underlying taxonomy that categorizes technical topics within the field. To achieve this, we propose a two-stage *taxonomy-then-generation* framework. In the first stage, a taxonomy tree is constructed, categorizing papers based on techniques and topics relevant to the query. In the second stage, detailed content is generated for each topic based on this taxonomy and the corresponding set of related papers. The quality of the review depends heavily on how well this taxonomy organizes the topics, enabling a coherent and comprehensive overview of the field.

This definition highlights several key challenges as follows:

***Challenge 1: How to accurately retrieve the relevant structured information from vast amount of citation networks?*** In LRG, the retrieval process is challenging due to the large size and complexity of citation networks, which contain not only textual information from individual papers but also

Figure 2: **HiReview (taxonomy-then-generation).** (a) Given a literature review topic, the most relevant community in the citation network is retrieved. (b) Papers are hierarchically divided into different clusters. (c) The central topic of each cluster at every level is generated. (d) Finally, the content of the literature review is generated with the hierarchical taxonomy.

structural relationships (i.e., citations). Successful LRG requires jointly considering the textual and structural information to identifying the most relevant sub-network from a vast citation network.

***Challenge 2: How can diversified documents be classified by identifying correlations between them, taking into account the synergy between their textual content and topological structure?*** The topology of the citation network plays a critical role in writing a literature review, as citations indicate the influence and relevance of papers within a research field. Papers with stronger connections in the citation network tend to be more closely related. Therefore, an effective LRG system must consider both the textual content of papers and their relationships within the network to accurately identify clusters of relevant literature and reveal underlying patterns.

***Challenge 3: How to generate the taxonomy in a hierarchical manner?*** The process of generating a coherent taxonomy tree is inherently hierarchical, with each level of the taxonomy tree closely related to the others. LRG requires identifying relevant technical topics and organizing them in a hierarchical fashion, capturing both broad categories and fine-grained subtopics.

## 4 METHOD

In this section, we present our *taxonomy-then-generation* framework (Fig. 2). First, we introduce a graph retrieval strategy to address *Challenge 1*, which aggregates neighbor relevance scores during retrieval (Section 4.1). Next, we propose an end-to-end hierarchical taxonomy tree generation model, consisting of hierarchical clustering and hierarchical generation. Specifically, we introduce a novel hierarchical graph clustering approach that considers relationships between nodes across different levels of the hierarchy to tackle *Challenge 2* (Section 4.2.1), categorizing papers in a citation network hierarchically. Then, in Section 4.2.2, we propose a bottom-up iterative generation approach to determine the central topic of each cluster at every hierarchical level, ultimately forming a taxonomy to address *Challenge 3*. Finally, we leverage the hierarchical taxonomy to guide the literature review generation process, producing high-quality, citation-aware literature reviews.

### 4.1 GRAPH CONTEXT-AWARE RETRIEVAL

Given a citation graph $G = (V, E, \{T_u\}_{u \in V})$ and a literature review topic $Q$, our objective is to identify the most relevant papers for the literature review. Different from common retrieval task, which usually only consider the textual similarity between query and source documents individually, to achieve the goal of finding related research works, we need to simultaneously consider the textual level relevance and the citation pattern between papers. To address this issue, we propose our unique textual subgraph retrieval method. Specifically, for each paper $u \in V$, we compute a relevance score $R(u, Q)$ between its title $t_u \in T_u$ and the query $Q$ using BM25 (a widely used retriever based on sparse retrieval). Rather than relying solely on the individual attributes of each paper, we enhance relevance scoring by incorporating information from the paper's neighbors in the citation graph as:

$$\tilde{R}(u, Q) = R(u, Q) + \sum_{v \in \mathcal{N}(u)} \alpha \cdot R(v, Q), \quad (2)$$

where $\mathcal{N}(u)$ denotes the set of neighbors of $u$ in $G$, and $\alpha$ is a pre-defined weighting factor that controls the influence of neighboring papers. We empirically find that aggregating the relevance scores

of neighbors leads to a significant improvement in retrieval accuracy. This approach is effective because the relevance of a paper's neighbors offers valuable contextual information that enhances its overall relevance to the topic. Further details are provided in Appendix A.3. The top-$k$ nodes with the highest aggregated scores are selected to form the subset $V'$. Subsequently, we construct the subgraph $G'(V', E', \{T_u\}_{u \in V'})$, where $V'$ is the set of selected nodes, $E' = \{(u, v) \in E \mid u, v \in V'\}$. We retain only the edges between the selected top-$k$ papers, thereby preserving the citation relationships among the most relevant papers.

## 4.2 HIERARCHICAL TAXONOMY GENERATION

In a literature review, each topic at every level corresponds to a set of papers, which aligns with a cluster in the citation graph. Therefore, we approach hierarchical taxonomy generation through hierarchical graph clustering, followed by generating a central topic for each cluster.

### 4.2.1 HIERARCHICAL CITATION GRAPH CLUSTERING

We define a clustering function $f(\cdot)$, which takes the graph $G_l$ and node features $X_{G_l}$ at level $l$ as input, and then produces clusters, i.e. $C_l = f(G_l, X_{G_l})$, where each cluster corresponds to a subgraph $G'_l \subset G_l$. We leverage a text encoder to convert the title of nodes to initialization text embedding as $X_{G'} = \text{LM}(\{t_u\}_{u \in V'})^2$. For each cluster at level $l$, treating it as a hyper-node in the $l + 1$ level, we construct the graph at level $l + 1$ and generate new node features through an aggregation function $g$, i.e., $G_{l+1} = g(f(G_l, X_{G_l}))$. The process continues recursively until a stopping criterion is met, i.e., no further meaningful clusters can be formed.

**Clustering Function** $f(\cdot)$. At each level $l$, we first update the node embeddings using a $\text{GNN}_\theta$, aggregating the features of neighboring nodes, as incorporating the information from references in the citation network is crucial. Then, we predict the probability $\hat{p}_{uv}$ that two nodes belong to the same cluster using an $\text{MLP}_\phi$ followed by a softmax transformation, where the concatenation of node embeddings for nodes $u$ and $v$ serves as the input. We then calculate the node density, which measures the similarity-weighted proportion of same-cluster nodes in its neighborhood as,

$$\hat{d}_u = \frac{1}{|\mathcal{N}(u)|} \sum_{k \in \mathcal{N}(u)} \hat{p}_{uk} \cdot \frac{h_u \cdot h_k}{\|h_u\|\|h_k\|}. \tag{3}$$

This density assesses how densely connected each node is within its local neighborhood. High-density nodes are more likely to be in the core of a cluster, whereas low-density nodes are more likely to be in ambiguous regions between clusters. $\hat{d}_u = 0$ if $u$ is isolated. Given $p_{uv}, \hat{d}_u, \hat{d}_v$, and a pre-defined edge connection threshold $p_\tau$, we update the candidate cluster as,

$$c_u = \{u\} + \{v \mid \hat{d}_u < \hat{d}_v \text{ and } p_{uv} > p_\tau\}, \tag{4}$$

where nodes with higher density are added to multiple candidate clusters. This approach avoids prematurely merging large clusters without clear boundaries. Early merging can obscure the nuanced relationships between papers, especially in citation networks where subtopics often overlap. *At the first level*, a soft clustering strategy is used, i.e., two clusters may overlap because one paper may be related to multiple topics in a literature review. We obtain the clusters in base level as,

$$C_1 = \{c_u \mid c_u \not\subset c_v\}, \ u \neq v \in V_1, \ |c_u| \neq 1. \tag{5}$$

This definition ensures that the final clusters capture distinct and relevant topics while still allowing for some overlap when a paper contributes to multiple topics. *At higher levels*, the clustering strategy transitions into hard clustering, as a single topic cannot belong to two different overarching topics in the taxonomy tree of the literature review. This implies that the clusters should form distinct connected components. Therefore, we generate the candidate edge set as,

$$\mathcal{E} = \{(u, v) \mid \underset{v \in c_u}{\arg\max}\, p_{uv}\}, \ u \in V_l, \ l > 1. \tag{6}$$

After a complete traversal of every node in $V_l$, $\mathcal{E}$ forms a set of connected components, resulting in disjoint clusters. This ensures that topics within the same cluster are densely connected, while maintaining clear boundaries between different topics.

**Aggregation Function** $g(\cdot)$. After the clustering function generates clusters $C_l = \{c_l^i\}_{i=1}^{|V_{l+1}|}$, we build up the graph $G_{l+1}(V_{l+1}, E_{l+1})$ at level $l + 1$ by serving the clusters as the hyper-nodes and

---

[2]Specifically, SentenceBert (Reimers, 2019) is used to generate the text embeddings.

linking these nodes with the cluster density. The feature of the hyper-node is defined as the aggregation of the average feature and representative feature of the corresponding cluster:

$$x_u = \frac{1}{|c_l^i|} \sum_{z \in c_l^i} h_z + h_k, \text{ where } k = \underset{z \in c_l^i}{\arg\max} \, \hat{d}_z, \tag{7}$$

$u \in V_{l+1}$ is the node corresponding to cluster $c_l^i$ at level $l+1$. This aggregation ensures that the hyper-node captures both the breadth (through the average) and the focus (through the distinctive feature) of the cluster. Given two hyper-nodes, $u$ and $v$, corresponding to clusters $c_l^i$ and $c_l^j$, an edge $e_{uv} \in E_{l+1}$ exists if any node in $c_l^i$ is linked to any node in $c_l^j$ at level $l$.

### 4.2.2 TAXONOMY TREE GENERATION

Each cluster at every level corresponds to a specific topic within the literature review, collectively forming the taxonomy tree of the literature review. To capture relationships within and between clusters and facilitate text generation from the underlying textual graph, we employ a $\text{PLM}_\Theta$ to generate the central topic for each cluster in a bottom-up manner, integrating both graph-based and text-based prompts. For a cluster $c_l^i$ at level $l$, we extract the corresponding subgraph and aggregate all relevant information into a graph embedding $h_{c_l^i}$ using $\text{GNN}_\theta$, then align its dimensions with the PLM's text vector through an $\text{MLP}_\Phi$, thereby integrating the relationships between topics into the PLM. For the node $u \in V_{l+1}$ that corresponds to a cluster $c_l^i$, the topic that node $u$ represents is generated as:

$$p_{\Theta,\theta,\phi,\Phi}(Y_{l+1}^u \mid c_l^i, q) = \prod_{i=1}^r p_{\Theta,\theta,\phi,\Phi}(y_i \mid y_{<i}, h_{c_l^i}, \{\mathcal{T}_l^j\}_{j \in c_l^i}, q), \tag{8}$$

where $q$ is the instruction to PLM, and $\mathcal{T}_l^j$ is the concatenation of the topic $Y_l^j$ and the titles of all papers under node $j \in V_l$. When $l = 1$, $\mathcal{T}_l^j$ is simply the title of node $j$, as node $j$ at the leaf level represents a single paper. After generating the central topic for each cluster at every level, we merge the topics across different levels to form the final taxonomy tree $\hat{Y}$.

### 4.3 CONTENT GENERATION

Once the taxonomy tree is established, we prompt the LLM, such as GPT-4o, to generate content for each topic in the taxonomy tree in parallel as $R_l^i = \text{Draft}(\tilde{Y}, Y_l^i, c_l^i)$. The LLM is provided with the complete taxonomy tree, the specific topic of focus, and the content of relevant papers within the cluster (Appendix A.6). This ensures that the LLM not only understands the topic and references required for generation, but also the hierarchical context of the topic. After generating content for all topics, they are merged to form the complete literature review $\mathcal{R}$. Multiple versions of the literature review are generated and evaluated by the LLM, which assess aspects such as content coverage and structure. Finally, the best version of the literature review is selected as the final output.

### 4.4 TRAINING STRATEGY

We jointly train the hierarchical clustering model and the topic generator to ensure that the hierarchical clusters formed are meaningful from a textual perspective, and that the topics generated are coherent with the structural information embedded in the citation network. The final objective function is given as,

$$\underset{\Theta,\theta,\phi,\Phi}{\arg\min} \mathcal{L} = \mathcal{L}_{\text{HiCluster}} + \mathcal{L}_{\text{PLM}}, \tag{9}$$

where $\mathcal{L}_{\text{HiCluster}}$ is the loss function for hierarchical clustering, and $\mathcal{L}_{\text{PLM}}$ is the loss function for topic generation. Since only a small portion of the citation network has been labeled for hierarchical clustering (as it is challenging to collect labels that indicate which hierarchical cluster the conferences in a literature review belong to), and the learning dynamics of GNNs and LLMs differ significantly, directly training both models simultaneously can result in a complex and unstable optimization process. This leads to the GNN struggling to learn meaningful clusters early on, which subsequently hinders the LLM's ability to generate coherent topics. Therefore, we pre-train the hierarchical clustering module to simplify the optimization process for the PLM, allowing the PLM to focus solely on content generation.

**To update** $\theta, \phi$. For hierarchical clustering, we train the GNN with two complementary objectives: the first optimizes clustering performance, while the second employs a hierarchical contrastive loss,

inspired by Zhang et al., to ensure that nodes belonging to the same cluster at different hierarchical levels are brought closer together in the embedding space:

$$\mathcal{L}_{\text{HiCluster}} = \mathcal{L}_{\text{cluster}} + \mathcal{L}_{\text{HiMulCon}} = \sum_{l \in L} \frac{-1}{|E|} \sum_{(u,v) \in E} q_{uv}^l \log \hat{p}_{uv}^l + (1 - q_{uv}^l) \log(1 - \hat{p}_{uv}^l)$$

$$+ \frac{1}{|L|} \sum_{l \in L} \sum_{u \in V} \frac{-\lambda_l}{|S_l(u)|} \sum_{s_l \in S_l(u)} \log \frac{\exp(\text{sim}(h_u, h_{s_l})/\tau)}{\sum_{k \in V \setminus u} \exp(\text{sim}(h_u, h_k)/\tau)}, \quad (10)$$

where $p_{uv}^l$ is the probability that two nodes belong to the same cluster at level $l$. Note that this probability is always calculated for leaf-level nodes, i.e., $u, v \in V$. At higher levels ($l > 1$), if $u$ and $v$ belong to two different clusters, $p_{uv}^l$ corresponds to the probability that these two hyper-nodes belong to the same cluster at level $l + 1$. $S_l(u)$ represents the set of positive samples for node $u$ at level $l$, i.e., nodes that belong to the same cluster. $\lambda_l$ is a weighting factor for the loss at level $l$, and $\tau$ is the temperature parameter. $L$ represents the total number of hierarchical levels.

**To update $\Theta, \Phi$.** After pre-training GNN$_\theta$, we fix $\theta, \phi$ and then jointly fine-tune GNN$_\theta$ and PLM$_\Theta$ to generate the central topic for each cluster. The MLP$_\Phi$ maps the graph embedding $h_{c_l^i}$ in Eq. 8 to the graph token embedding $\mathbf{h}_c \in \mathbb{R}^{d_{\text{LLM}}}$. We leverage the text embedder function of the LLM to convert them into text embeddings $\mathbf{h}_t \in \mathbb{R}^{L_t \times d_{\text{LLM}}}$, where $L_t$ indicates the token length of the concatenated text contexts. Finally, the generation process in Eq. 8 becomes:.

$$p_{\Theta, \theta, \phi, \Phi}(Y_{l+1}^u \mid c_l^i, q) = \prod_{i=1}^r p_{\Theta, \Theta, \phi, \Phi}(y_i \mid y_{<i}, [\mathbf{h}_c; \mathbf{h}_t]). \quad (11)$$

The fine-tuning of PLM$_\Theta$ is conducted using Low-Rank Adaptation (LoRA) (Hu et al., 2021), with MLP$_\Phi$ serving as an adapter for graphs, enabling efficient adjustment of both models.

## 5 EXPERIMENT

To demonstrate the improvement of the proposed taxonomy-then-generation method on the generation of literature review papers, we conducted comprehensive experiments. In addition to analyzing the generative quality of the literature review, this section will answer the following questions:

- **RQ1.** Why use hierarchical clustering instead of a simple clustering strategy?
- **RQ2.** Why do we need a taxonomy tree to guide the generation process?
- **RQ3.** Why not prompt LLMs to generate taxonomy trees and then literature reviews?

**Setup.** The fine-tuned topic generator is LLaMA (Touvron et al., 2023), while GPT-4o serves as the content generator[3]. For hierarchical clustering, we employ the GAT (Veličković et al., 2017) in hierarchical clustering. Implementation details can be found in Appendix A.2.

### 5.1 DATASET

We manually collected 518 high-quality literature review articles with clear taxonomy or well-defined article structures from arXiv[4]. Most of these review papers were published within the past three years. For each literature review, we extracted its taxonomy tree and gathered its 2-hop citation network, which includes the direct citations of the review paper and the citations of the review's references. The mutual citation relationships among these references form a complex citation network for each literature review. These 2-hop citation networks contain an average of 6,658.4 papers and 11,632.9 edges, accommodating isolated papers. More details are in Appendix A.1.

### 5.2 EVALUATION

We compare the reviews generated by our model with those written by human experts, zero-shot LLMs and naive RAG-based LLMs, i.e., GPT-4o and Claude-3.5. While zero-shot LLMs rely solely on their pretrained knowledge to generate literature review content, naive RAG-based LLMs utilize a simple BM25 retriever. The LLM backbone of AutoSurvey and HiReview are both GPT-4o. Additionally, we benchmarked our model against the state-of-the-art review generator, AutoSurvey

---

[3]Specifically, we use LLaMA-2-7b and gpt-4o-2024-05-13.
[4]https://arxiv.org/

Table 1: Results of literature review generation by pure LLMs, naive RAG-based LLMs, AutoSurvey and HiReview. The best performance is in **Bold**. LLM$^\circ$ indicates the LLM is provided top-500 relevant papers retrieved by naive BM25.↑ indicates that a higher metric value corresponds to better model performance.

| Model | LLMScores↑ | | | | BertScore ↑ |
|---|---|---|---|---|---|
| | Coverage ↑ | Structure ↑ | Relevance ↑ | Average ↑ | |
| **Human-written** | $1._{0000}$ | $1._{0000}$ | $1._{0000}$ | $1._{0000}$ | $1._{0000}$ |
| Pure LLMs | | | | | |
| **GPT-4o** | $0.7430_{\pm 0.12}$ | $0.8346_{\pm 0.11}$ | $0.8225_{\pm 0.07}$ | $0.8000$ | $0.8127_{\pm 0.03}$ |
| **Claude-3.5** | $0.7224_{\pm 0.09}$ | $0.8116_{\pm 0.14}$ | $0.7948_{\pm 0.09}$ | $0.7763$ | $0.8130_{\pm 0.04}$ |
| Naive RAG-based LLMs | | | | | |
| **GPT-4o$^\circ$** | $0.8219_{\pm 0.13}$ | $0.8293_{\pm 0.12}$ | $0.8972_{\pm 0.06}$ | $0.8495$ | $0.8094_{\pm 0.03}$ |
| **Claude-3.5$^\circ$** | $0.8339_{\pm 0.11}$ | $0.8215_{\pm 0.13}$ | $0.9051_{\pm 0.05}$ | $0.8535$ | $0.8141_{\pm 0.02}$ |
| Auto Review System | | | | | |
| **AutoSurvey** | $0.8646_{\pm 0.07}$ | $0.9122_{\pm 0.05}$ | $0.9093_{\pm 0.04}$ | $0.8957$ | $0.8256_{\pm 0.02}$ |
| **HiReview** | $\mathbf{0.9163}_{\pm 0.03}$ | $\mathbf{0.9484}_{\pm 0.02}$ | $\mathbf{0.9428}_{\pm 0.01}$ | $\mathbf{0.9358}$ | $\mathbf{0.8449}_{\pm 0.02}$ |

(Wang et al., 2024). We evaluate the quality of generated content with LLMScore and BertScore (Zhang et al., 2019). For LLMScore, Wang et al. demonstrated that LLM-based evaluations of literature review align well with human preferences. Therefore, we employed multiple LLMs to assess the overall quality of the generated content and averaged their evaluation scores. Following AutoSurvey's scoring criteria, we evaluate the generated reviews based on coverage, structure, and relevance when selecting the best output and calculating the LLMScore. When evaluate the review content, instead of directly scoring the generation, we have LLMs compare the generated content with human-written reviews (Appendix A.6).

**Main Results.** As shown in Table 1, Our method HiReview consistently outperforms the other review generation methods in all metrics. It excels across all LLMScore categories, with notably high structure (0.9484) and relevance (0.9428) scores. AutoSurvey employs a structured methodology that combines retrieval, outline generation, and section drafting, leading to superior content generation compared to naive systems (with average LLMScore of 0.8957).

Pure LLMs and naive RAG-based LLMs struggle with both stability and performance, which makes them unreliable for consistent literature review generation. AutoSurvey reduces this instability through prompt design and multi-output generation, achieving *Structure* ±0.05 and *Relevance* ±0.04—lower deviations than those of pure and naive RAG-based LLMs. HiReview, however, outperforms all other models across all metrics, with consistently low standard deviations. This demonstrates HiReview's superior stability and consistency in generating high-quality reviews (**RQ3**). Its success can be attributed not only to HiReview's use of a graph-context-aware retrieval method but also to the taxonomy tree, which provides hierarchical context for domain-specific concerns within the large language model. A detailed analysis contrasting the *outline-then-generation* with the *taxonomy-then-generation*, based on a specific generation example, is provided in Appendix A.5. Furthermore, an example of the generation for a cluster with the central topic *Continual Text Classification* is included in Appendix A.7.

## 5.3 ABLATION STUDY

Although we demonstrate the performance of HiReview (taxonomy-then-generation) in terms of the quality of the literature review generated, we will assess the impact of various components on the performance of HiReview.

**Impacts of Components.** As shown in Table 2, we test three variants of HiReiview mode. *HiReview w/o retrieval* refers to the variant where the graph retrieval module is removed, and all papers in the citation network are used. A significant drop is observed across all metrics, particularly in

Table 2: Ablation study results for HiReview.

| Model | LLMScores↑ | | |
|---|---|---|---|
| | Coverage ↑ | Structure ↑ | Relevance ↑ |
| **HiReview** | 0.9163 | 0.9484 | 0.9428 |
| **w/o retrieval** | 0.6705 | 0.7216 | 0.7073 |
| **w/o clustering$^*$** | 0.8863 | 0.9261 | 0.9314 |
| **w/o taxonomy** | 0.8612 | 0.8790 | 0.9078 |

*Coverage* (0.9163 → 0.6705) and *Relevance* (0.9428 → 0.7073). This indicates that the inclusion of unrelated papers introduces substantial noise, negatively impacting both the taxonomy tree generation (due to an excess of negative samples in hierarchical clustering) and content generation (where the noise hinders the creation of precise summaries). As a result, the quality opf generated summaries is even worse than those produced by zero-shot LLMs.

*HiReview w/o clustering** bypasses the clustering process and directly uses the retrieved papers to generate the taxonomy. Instead of iteratively generating topics at each level, this variant creates the taxonomy in a single step. It is marked with $*$ because the topic generator in this case is an LLM i.e., GPT-4o, rather than a fine-tuned LLaMA, as the number of taxonomy trees is insufficient for effective fine-tuning. Although this variant performs worse than HiReview, it still delivers competitive performance, outperforming naive RAG-based LLMs and AutoSurvey. This suggests that the combination of the graph retrieval module and the taxonomy-then-generation paradigm is more effective than naive retrieval-then-generation and outline-then-generation approaches.

*HiReview w/o taxonomy* removes the hierarchical taxonomy tree generation module, instead using paper clusters to prompt the LLM for both topic generation and content creation. The absence of a hierarchical taxonomy reduces the model's ability to leverage topic relations across different levels, leading to less organized and relevant content (*Structure*: $0.9484 \rightarrow 0.8790$ and *Coverage*: $0.9163 \rightarrow 0.8612$). Similar to *HiReview w/o clustering**, *HiReview w/o taxonomy* does not use fine-tuned LLaMA, and its performance is more degraded. This indicates that when using pure LLM generation methods, generating a hierarchical taxonomy tree to guide content generation significantly enhances the quality of the output. Finally, we can answer remaining questions raised at the beginning of the Experiment Section.

**RQ1.** We consider two baseline methods for the hierarchical clustering module: one that utilizes an LLM (i.e., GPT-4o) to cluster papers and another that applies $K$-means, adjusting the number of clusters to represent different

Table 3: Accuracy of hierarchical clustering.

| Model | Level 1 | Level 2 | Average |
|---|---|---|---|
| **HiCluserting** | 0.7127 | 0.6395 | 0.6761 |
| **K-means** | 0.3723 | 0.4201 | 0.3962 |
| **LLM clustering** | 0.4296 | 0.4518 | 0.4407 |

levels. However, neither method can be jointly trained with the topic generator. Even disregarding the training requirement, the hierarchical nature of the literature review's taxonomy tree requires soft clustering at the initial layer and hard clustering at subsequent layers—an issue that no existing work addresses. As shown in Table 3, when considering the clustering task alone, both baselines underperform compared to our hierarchical approach. Although the LLM is prompted to perform soft clustering at the first level, offering a slight improvement over $K$-means, it still does not achieve the effectiveness of our hierarchical clustering approach.

**RQ2.** As shown in Table 1, HiReview, which incorporates a taxonomy tree, outperforms all other models, particularly in *Structure*, achieving a score of 0.9484. In contrast, AutoSurvey, which follows an outline-then-generation approach without hierarchical taxonomy, shows lower scores, such as a *Structure* score of 0.9122. The ablation study further supports this. When the taxonomy is removed, the structure score drops significantly (as in Table 2). This demonstrates that the taxonomy tree plays a critical role in organizing and guiding the content generation process, especially when maintaining a clear structure is crucial for a literature review. The taxonomy ensures more coherent and relevant summaries. Without providing the taxonomy tree, the generation loses its hierarchical guidance, leading to less structured and less comprehensive content.

## 6 CONCLUSION

In this paper, we propose HiReview, a novel method for literature review generation that leverages a hierarchical taxonomy-driven approach, integrating graph-based clustering with graph context-aware retrieval-augmented large language models. Extensive experiments show that HiReview consistently outperforms state-of-the-art methods across multiple metrics. The ablation study provides strong evidence of the critical role that the hierarchical taxonomy tree plays in guiding the content generation process, leading to more coherent and comprehensive reviews. Compared to the outline-then-generation framework, the taxonomy-then-generation approach offers improved robustness and performance. HiReview achieves a more structured and contextually aligned synthesis of scientific literature than traditional outline-based methods. This work addresses key challenges in automatic literature review generation, particularly in managing large citation networks and preserving the structural integrity of scientific literature reviews. The results establish a new benchmark in the field and emphasize the importance of incorporating both structural and content-based guidance in LLM-driven review generation.

ACKNOWLEDGMENTS

We thank all the researchers in the community for producing high-quality literature review papers. These articles are the basis of this study.

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

## A APPENDIX

### A.1 DATASET

We conduct experiments on 2-hop citation networks for each literature review paper rather than randomly collecting papers to construct a large citation network as a database. This is because using a large, random network complicates performance evaluation, making it difficult to assess both retrieval and clustering accuracy. Additionally, if related papers published after the literature review are present in the citation network, the retriever may include these newer papers in generating the review content. This would lead to an unfair comparison when evaluate the generated content against the human-written review.

**Citation Network Construction Process.** For each literature review, we first extracted its references and constructed a citation tree, with the review paper as the root and its cited papers as the leaves. We then repeated this process for each cited paper, constructing a citation tree for each one. Next, we merged all these trees into a single, large citation network, consolidating any duplicate nodes. To automate this process, we used citation information from arXiv, which provides the La-TeX source code for each paper, including the bib or bbl files. If a paper was available on arXiv, we extracted its .tex file to obtain both the abstract and full text, using these as high-quality text features for the corresponding node. We used the arXiv API to automate this process. For papers not available on arXiv, we used the Google Scholar API to automatically retrieve the abstract, which we used as the text feature for the corresponding node in the citation network. Finally, we removed the node representing the original literature review, leaving 1-hop, 2-hop, and 3-hop citation networks for each review.

### A.2 IMPLEMENTATION

All experiments were conducted on a Linux-based server equipped with 4 NVIDIA A10G GPUs. For 518 review papers, we successfully collected taxonomy trees with hierarchical clustering labels for 313 of the literature reviews. Of these, 200 reviews were used to train the hierarchical clustering and taxonomy generation module, while the remaining 118 were used to test the performance of the pre-trained hierarchical clustering model. 318 reviews were reserved for a comprehensive evaluation of review content generation. The number of articles retrieved in retrieval phase was set to 200. The scaling factor $\alpha$ is set to 1. The code and dataset are available at https://anonymous.4open.science/r/HiReivew-767D.

**Pre-Train Hierarchical Clustering Module.** GNN used in this paper is GAT (Veličković et al., 2017) which has 2 layers with 4 heads per layer and a hidden dimension size of 1024. MLP$_\phi$ has 2 layers and a hidden dimension size of 1024. The edge connection threshold $p_\tau$ is searched in [0.1, 0.2, 0.5, 0.8]. The clustering model is trained for a maximum of 500 epochs using an early stop scheme with patience of 10. The learning rate is set to 0.001. The training batch is set to 512 and the test batch is 1024.

**Fine-Tuning.** The LLM backbone is Llama-2-7b-hf. We adopt Low Rank Adaptation (LoRA) (Hu et al., 2021) for fine-tuning, and configure the LoRA parameters as follows: the dimension of the low-rank matrices is set to 8; the scaling factor is 16; the dropout rate is 0.05. For optimization, the AdamW optimizer is used. The initial learning rate is set to 1e-5 and the weight decay is 0.05. Each experiment is run for a maximum of 10 epochs, with a batch size of 4 for both training and testing. The MLP$_\Phi$ has 2 layers and a hidden dimension size of 1024.

**LLMs.** When calling the API, we set temperature as 1 and other parameters to default. The content generator is gpt-4o-2024-05-13 and the content judge is and claude-3-haiku-20240307.

### A.3 INVESTIGATION OF RETRIEVAL MODELS

We experimented with different retrieval models and strategies, testing two representative methods: the sparse retrieval model, BM25 (Robertson et al., 2009), and the dense retrieval model, Sentence-Bert (Reimers, 2019). In citation networks, neighbor information and the topological structure play a crucial role in retrieval, as papers on the same topic often cite each other. To assess the impact of using neighbor information, we applied two retrieval strategies for both models: one incorporating neighbor information as described in Section 4.1 (*Retrieval w/ Neighbor*) and the other excluding neighbor information (*Retrieval w/o Neighbor*). Given a topic (specifically, the title of a review pa-

per), we retrieved papers related to this topic from the citation network and measured the accuracy by calculating how many of the retrieved papers appeared in the references of the corresponding literature review. The number of retrieved papers was not fixed, but matched the reference count for each review paper.

Table 4: Results of retrieval on the citation network corresponding to 50 review papers. *2-hop* and *3-hop* represent citation networks of review papers at different scales. *1-hop (merged)* refers to the 1-hop citation network of a review paper, merged with all other 1-hop citation networks, different review papers. Similarly, *2-hop (merged)* is constructed by merging the 2-hop citation network of a review with all other 49 review papers.

| Model | Accuracy↑ | | | |
|---|---|---|---|---|
| | 1-hop (merged) | 2-hop | 2-hop (merged) | 3-hop |
| Retrieval w/o Neighbor | | | | |
| **BM25** | 0.3308 | 0.1375 | 0.0947 | 0.1014 |
| **SentenceBert** | 0.5234 | 0.1746 | 0.1521 | 0.1490 |
| Retrieval w Neighbor | | | | |
| **BM25** | 0.7445 | 0.6435 | 0.5950 | 0.6179 |
| **SentenceBert** | 0.2602 | 0.2758 | 0.2181 | 0.2144 |

As shown in Table 4, SentenceBert consistently outperforms BM25 across all scales when neighbor information is not used. For example, in the *1-hop merged* case, SentenceBert achieves an accuracy of 0.5234, significantly higher than BM25's 0.3308. However, both methods show relatively low accuracy without neighbor information, and their performance declines as the size of citation networks increases, indicating that retrieving relevant papers becomes more challenging as the network expands. In contrast, BM25 significantly outperforms SentenceBert when neighbor information is utilized. For instance, in the *1-hop merged* case, BM25 reaches an accuracy of 0.7445, while SentenceBert's accuracy drops sharply to 0.2602. BM25 maintains much higher accuracy across all scales with neighbor information. BM25, as a sparse retrieval model, relies on exact term matches, which is particularly advantageous in structured environments like citation networks, where specific terms (e.g., paper titles or keywords) are highly relevant. The inclusion of neighbor information allows BM25 to better capture relationships between papers by focusing on direct term matches in titles or citations. When neighbor information is introduced, the context around the target paper becomes more critical. BM25 effectively leverages this by prioritizing exact matches from neighboring papers, while SentenceBert, which focuses on semantic similarity, may lose precision when handling a broader context that includes less directly related papers.

Without graph-aware retrieval, methods like AutoSurvey must retrieve a large number of papers (e.g., 1200 in AutoSurvey) to avoid missing relevant ones. Retrieving fewer papers risks missing important content, while retrieving too many introduces noise from irrelevant papers. Graph-aware retrieval significantly alleviates this issue. The graph context-aware retrieval strategy we propose achieves more accurate results with fewer retrievals, i.e., 200, reducing irrelevant information and contributing to the superior generation performance of our model. Moreover, even when applied to large citation networks (such as *2-hop merged* each containing over 200,000 papers), our method maintains stable retrieval accuracy, demonstrating HiReview's robustness across different citation network sizes. Additionally, we experimented with different retrieval strategies, such as retrieving based on both the title and abstract. We found that using only the title yielded the best results, while incorporating additional information like the abstract reduced retrieval performance.

### A.4 THE CHOICE OF GNN

In addition to GAT (Veličković et al., 2017), we also explored other GNNs as graph encoders, i.e., GCN (Kipf & Welling, 2016) and Graph Transformer (Shi et al., 2020). The comparison results of these models on clustering are shown in Table 5.

Table 5: Performance of different GNN on hierarchical clustering.

| Model | Level 1 | Level 2 | Average |
|---|---|---|---|
| **GAT** | 0.7127 | 0.6395 | 0.6761 |
| **GCN** | 0.6730 | 0.5963 | 0.6347 |
| **Graph Transformer** | 0.6811 | 0.6024 | 0.6418 |

GAT achieves the highest performance across both levels, with an average score of 0.6761. It outperforms the other models at both Level 1 (0.7127) and Level 2 (0.6395), making it the most effective GNN for this task. This superior performance can likely be attributed to GAT's attention mechanism, which enables the model to assign varying importance weights to neighboring papers, allowing it

to better capture the hierarchical structure of the graph. As a result, we selected GAT as the GNN backbone for HiReview.

## A.5 COMPARISON TO OUTLINE-THEN-GENERATION

Recently, the paradigm for generating literature reviews has shifted from extractive models to outline-based generative models, where LLMs are used to generate an outline of the literature review and then the outline is used to guide generation of content (i.e., *outline-then-generation* in (Wang et al., 2024; Wu et al., 2024)). Taking *Lifelong Learning of Large Language Models* as an example topic, we demonstrate the generation of HiReview using two different paradigms: *outline-then-generation* and *taxonomy-then-generation*. We analyze the gains and losses of each approach. For the *outline-then-generation* paradigm, we allow the LLM to generate the outline based on clusters at the base level.

**Comparison.** As illustrated in Fig. 4 and Fig. 5, both the human-designed taxonomy and the taxonomy generated by HiReview for Lifelong Learning of LLMs provide a clear and cohesive hierarchical structure. In contrast, the outline in Fig. 3 covers plausible approaches and concepts related to lifelong learning and organizes these elements. The outline provides a broad structure covering everything from basics to future trends. For instance, it includes *Introduction* and *Future Directions and Emerging Trends*, allowing for an overview of the literature review. The taxonomy clearly outlines specific applications like *Continual Relation Extraction*, allowing for a more focused discussion on particular areas of lifelong learning. Therefore, use outline-then-generation for comprehensive, structured reviews that cover both theoretical and practical aspects, making it particularly effective for diverse audiences. In contrast, use taxonomy-then-generation for more focused, task-specific reviews, especially when the emphasis is on core concepts, and the audience is already familiar with the basics of a specific domain.

**When to use *outline-then-generation*?** The outline offers a logical and detailed structure that ensures comprehensive coverage of all important aspects of the topic. It follows a clear progression from introduction to conclusion, making it easier for readers to follow. This approach includes broad coverage, addressing theoretical foundations, architectures, training methods, applications, challenges, evaluation methods, and future trends. However, it places less emphasis on specific lifelong learning scenarios, as the outline doesn't explicitly highlight particular tasks or domains where lifelong learning is applied. The *outline-then-generation* paradigm is ideal when a structured, logically flowing review is needed, covering the topic comprehensively from foundational principles to advanced applications.

**When to use *taxonomy-then-generation*?** The taxonomy clearly highlights different domains and tasks where lifelong learning is applied (e.g., relation extraction, semantic segmentation). It emphasizes core lifelong learning concepts, directly addressing key issues like catastrophic forgetting and lifelong learning strategies. This approach offers flexibility, allowing for easy addition of new categories or tasks as the field evolves. It also provides a concise overview, offering a quick snapshot of the main areas of research in lifelong learning for LLMs. However, it may be less comprehensive, missing broader contexts such as theoretical foundations, evaluation methods, and future directions. Additionally, it doesn't provide a narrative flow from basic concepts to advanced applications. The *taxonomy-then-generation* paradigm is best suited for highlighting specific areas, offering flexible categorization, and providing a concise overview of research without requiring a detailed or linear narrative.

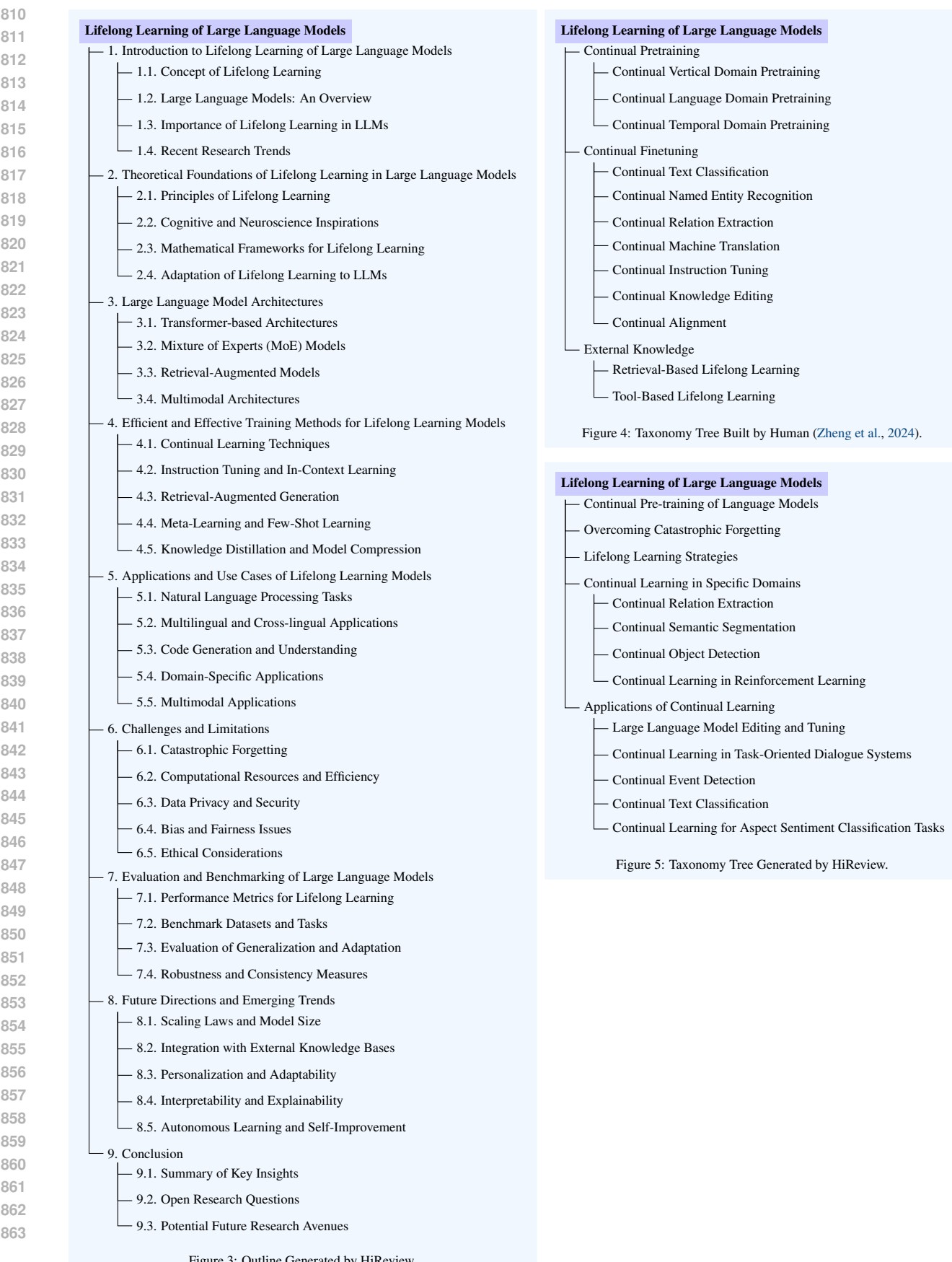

**Lifelong Learning of Large Language Models**

— 1. Introduction to Lifelong Learning of Large Language Models
— 1.1. Concept of Lifelong Learning
— 1.2. Large Language Models: An Overview
— 1.3. Importance of Lifelong Learning in LLMs
— 1.4. Recent Research Trends
— 2. Theoretical Foundations of Lifelong Learning in Large Language Models
— 2.1. Principles of Lifelong Learning
— 2.2. Cognitive and Neuroscience Inspirations
— 2.3. Mathematical Frameworks for Lifelong Learning
— 2.4. Adaptation of Lifelong Learning to LLMs
— 3. Large Language Model Architectures
— 3.1. Transformer-based Architectures
— 3.2. Mixture of Experts (MoE) Models
— 3.3. Retrieval-Augmented Models
— 3.4. Multimodal Architectures
— 4. Efficient and Effective Training Methods for Lifelong Learning Models
— 4.1. Continual Learning Techniques
— 4.2. Instruction Tuning and In-Context Learning
— 4.3. Retrieval-Augmented Generation
— 4.4. Meta-Learning and Few-Shot Learning
— 4.5. Knowledge Distillation and Model Compression
— 5. Applications and Use Cases of Lifelong Learning Models
— 5.1. Natural Language Processing Tasks
— 5.2. Multilingual and Cross-lingual Applications
— 5.3. Code Generation and Understanding
— 5.4. Domain-Specific Applications
— 5.5. Multimodal Applications
— 6. Challenges and Limitations
— 6.1. Catastrophic Forgetting
— 6.2. Computational Resources and Efficiency
— 6.3. Data Privacy and Security
— 6.4. Bias and Fairness Issues
— 6.5. Ethical Considerations
— 7. Evaluation and Benchmarking of Large Language Models
— 7.1. Performance Metrics for Lifelong Learning
— 7.2. Benchmark Datasets and Tasks
— 7.3. Evaluation of Generalization and Adaptation
— 7.4. Robustness and Consistency Measures
— 8. Future Directions and Emerging Trends
— 8.1. Scaling Laws and Model Size
— 8.2. Integration with External Knowledge Bases
— 8.3. Personalization and Adaptability
— 8.4. Interpretability and Explainability
— 8.5. Autonomous Learning and Self-Improvement
— 9. Conclusion
— 9.1. Summary of Key Insights
— 9.2. Open Research Questions
— 9.3. Potential Future Research Avenues

Figure 3: Outline Generated by HiReview.

**Lifelong Learning of Large Language Models**

— Continual Pretraining
— Continual Vertical Domain Pretraining
— Continual Language Domain Pretraining
— Continual Temporal Domain Pretraining
— Continual Finetuning
— Continual Text Classification
— Continual Named Entity Recognition
— Continual Relation Extraction
— Continual Machine Translation
— Continual Instruction Tuning
— Continual Knowledge Editing
— Continual Alignment
— External Knowledge
— Retrieval-Based Lifelong Learning
— Tool-Based Lifelong Learning

Figure 4: Taxonomy Tree Built by Human (Zheng et al., 2024).

**Lifelong Learning of Large Language Models**

— Continual Pre-training of Language Models
— Overcoming Catastrophic Forgetting
— Lifelong Learning Strategies
— Continual Learning in Specific Domains
— Continual Relation Extraction
— Continual Semantic Segmentation
— Continual Object Detection
— Continual Learning in Reinforcement Learning
— Applications of Continual Learning
— Large Language Model Editing and Tuning
— Continual Learning in Task-Oriented Dialogue Systems
— Continual Event Detection
— Continual Text Classification
— Continual Learning for Aspect Sentiment Classification Tasks

Figure 5: Taxonomy Tree Generated by HiReview.

## A.6 PROMPT USED

**Prompt for generating content of each cluster.**

Instruction: You are writing an overall and comprehensive literature review about **[TOPIC]**.

---

The taxonomy tree of this literature review is:
**[OVERALL TAXONOMY TREE]**

Now, you need to write the content for a section: **[TOPIC OF CLUSTER]**. The following is a list of references:

**[TITLE 1]: [CONTENT 1]**
**[TITLE 2]: [CONTENT 2]**
...
**[TITLE N]: [CONTENT N]**

Requirements:

- The subsection should contain more than **[WORD NUM]** words.
- When writing sentences based on specific papers, cite the paper using a numbered reference format [X], where X is the number corresponding to the paper in the reference list at the end of the document. The full titles of the papers should be listed in this reference section.

Citation Guidelines:

- Summarizing Research: Cite sources when summarizing existing literature.
- Using Specific Concepts or Data: Provide citations when discussing specific theories, models, or data.
- Using Established Methods: Cite the creators of methodologies you employ in your literature review.
- Supporting Arguments: Cite sources that back up your conclusions and arguments.

Only return the content with more than **[WORD NUM]** words that you write for the section **[TOPIC OF CLUSTER]** without any other information:

**Prompt for evaluating the coverage of generated review.**

**Instruction:** You are an expert in literature review evaluation, tasked with comparing a generated literature review to a human-written literature review on the topic of **[TOPIC]**.

---

Human-Written Literature Review (Gold Standard):
**[GROUND TRUTH REVIEW]**

Generated Literature Review (To be evaluated):
**[GENERATED REVIEW]**

---

**Evaluation Requirements:** The human-written literature review serves as the gold standard. Your job is to assess how well the generated literature review compares in terms of coverage. Carefully analyze both reviews and provide a score.

**Evaluate Coverage (Score out of 100).** Assess how comprehensively the generated review covers the content from the human-written review. Consider:

- The percentage of key subtopics addressed from the human-written review.
- The depth of discussion for each subtopic compared to the human-written version.
- Balance between different areas within the topic as presented in the human-written review.

Only return only a numerical score out of 100, where 100 represents perfect alignment with the human-written literature review, without providing any additional information.

---

Prompt for evaluating the structure of generated review.

**Instruction:** You are an expert in literature review evaluation, tasked with comparing a generated literature review to a human-written literature review on the topic of **[TOPIC]**.

---

Human-Written Literature Review (Gold Standard):

**[GROUND TRUTH REVIEW]**

Generated Literature Review (To be evaluated):
**[GENERATED REVIEW]**

---

**Evaluation Requirements:** The human-written literature review serves as the gold standard. Your job is to assess how well the generated literature review compares in terms of structure. Carefully analyze both reviews and provide a score.

**Evaluate Structure (Score out of 100).** Assess how well the generated literature review's organization and flow match that of the human-written literature review. Consider:

- Similarity in logical progression of ideas.
- Presence of a clear hierarchy of sections and subsections comparable to the human-written literature review.
- Appropriate use of headings and subheadings in line with the human-written version.
- Overall coherence within and between sections relative to the human-written literature review.

Only return only a numerical score out of 100, where 100 represents perfect alignment with the human-written literature review, without providing any additional information.

---

Prompt for evaluating the relevance of generated review.

**Instruction:** You are an expert in literature review evaluation, tasked with comparing a generated literature review to a human-written literature review on the topic of **[TOPIC]**.

---

Human-Written Literature Review (Gold Standard):

**[GROUND TRUTH REVIEW]**

Generated Literature Review (To be evaluated):
**[GENERATED REVIEW]**

---

**Evaluation Requirements:** The human-written literature review serves as the gold standard. Your job is to assess how well the generated literature review compares in terms of relevance. Carefully analyze both reviews and provide a score.

**Evaluate Relevance (Score out of 100).** Evaluate how well the generated literature review aligns with the focus and content of the human-written literature review. Consider:

- Alignment with the core aspects of **[TOPIC]** as presented in the human-written literature review.

- Relevance of examples and case studies compared to those in the human-written literature review.

- Appropriateness for the target audience as demonstrated by the human-written literature review.

- Exclusion of tangential or unnecessary information not present in the human-written version.

Only return only a numerical score out of 100, where 100 represents perfect alignment with the human-written literature review, without providing any additional information.

## A.7 EXAMPLE OF GENERATION

**Example Generation for Continual Text Classification.**

Continual Text Classification is a crucial area of research in lifelong learning for large language models, addressing the challenge of adapting to new categories and evolving data distributions over time. This field is particularly relevant in real-world applications where new topics or intents may emerge, such as during global events that introduce novel categories. The techniques employed in continual text classification can be broadly categorized into several approaches, including distillation-based methods, replay-based methods, regularization-based methods, and architecture-based methods.

Distillation-based methods focus on transferring knowledge from previous models to new ones. Some approaches use contrastive ensemble distillation to enhance the discriminability between text categories [1], while others employ multi-strategy rebalancing to tackle class imbalance issues [2]. Replay-based methods involve revisiting previously learned information to maintain performance on older tasks. These methods often integrate contrastive learning techniques to enhance the quality of replay samples [3] or use structured learning methods to improve the stability of the learning process [4].

Regularization-based techniques aim to constrain the model's parameter updates to prevent forgetting. Some approaches focus on selectively updating a small subset of parameters to reduce computational burden [5], while others implement adaptive regularization processes that automatically adjust to balance retaining old knowledge with acquiring new information [6]. Architecture-based methods modify the model structure to accommodate new tasks, with some proposing hierarchical overlay projections to optimize knowledge transfer across tasks [7] and others introducing mechanisms for parameter isolation to minimize interference between new and old tasks [8].

In addition to these primary approaches, researchers have explored data augmentation and generation techniques, using large language models to generate diverse and relevant samples for memory augmentation [9]. Meta-learning frameworks, although less common in text classification, have also been employed to enable quick adaptation to new tasks while mitigating catastrophic forgetting [10].

The application of these techniques has been explored in specific domains, such as medical intent detection [11] and social media content analysis [12]. As the field progresses, there is an increasing focus on developing methods that can handle few-shot scenarios [13] and multilingual settings [14]. These advancements aim to make continual text classification more practical and adaptable in real-world applications where data may be scarce or linguistically diverse, ultimately contributing to the development of more robust and versatile large language models capable of continuous learning and adaptation.

[1] CLASSIC: Continual and Contrastive Learning of Aspect Sentiment Classification Tasks
[2] Lifelong intent detection via multi-strategy rebalancing
[3] InfoCL: Alleviating Catastrophic Forgetting in Continual Text Classification from An Information Theoretic Perspective
[4] Class Lifelong Learning for Intent Detection via Structure Consolidation Networks
[5] Parameter-efficient Continual Learning Framework in Industrial Real-time Text Classification System
[6] Hyperparameter-free Continuous Learning for Domain Classification in Natural Language Understanding
[7] HOP to the Next Tasks and Domains for Continual Learning in NLP
[8] Prompts Can Play Lottery Tickets Well: Achieving Lifelong Information Extraction via Lottery Prompt Tuning
[9] Making Pre-trained Language Models Better Continual Few-Shot Relation Extractors
[10] Meta-Learning Improves Lifelong Relation Extraction
[11] Incremental intent detection for medical domain with contrast replay networks
[12] Lifelong Learning of Hate Speech Classification on Social Media
[13] Continual few-shot intent detection
[14] Learning to solve NLP tasks in an incremental number of languages

