# OpenReview forum: "HiReview: Hierarchical Taxonomy-Driven Automatic Literature Review Generation"
_ICLR.cc/2025/Conference — Submitted to ICLR 2025_

### Official Review · Reviewer_TsMV · 2024-10-24

**Soundness:** 2
**Presentation:** 2
**Contribution:** 2
**Rating:** 3
**Confidence:** 5

**Summary:**

The paper proposes a taxonomy-then-generation framework for literature review generation. In the clustering phase, a hierarchical clustering method is employed, and the final review is generated based on the hierarchical clustering results. The framework’s effectiveness is validated on the constructed dataset.

**Strengths:**

The taxonomy-then-generation approach is an effective strategy for generating long literature reviews. The proposed method achieves good results.

**Weaknesses:**

Shortcomings:
1. The experimental section does not compare the clustering method with that of GraphRAG.
2. There is a lack of important references: “Hierarchical Catalogue Generation for Literature Review” (EMNLP 2023 Findings), which is a highly relevant prior work but has not been considered.

Suggestions:
1. Many claims need citations for support. For example, in lines 031-041: “However, manual literature reviews are labor-intensive and time-consuming, especially when addressing fields with complex, hierarchical topics.” This should be substantiated by references. Similarly, in lines 045-047: “Recently, large language models (LLMs) have been applied to this task due to their ability to manage long-range dependencies and provide richer contextual understanding.” Relevant supporting work is missing.
2. Do not arbitrarily modify the paper template, such as the font size of the image captions.
3. Use consistent tenses. For example, line 052 states, “AutoSurvey (Wang et al., 2024) addresses,” but line 055 reads, “Similarly, Wu et al. introduced.”
4. When citing multiple references consecutively, it is better to list them in chronological order, such as in line 142, starting with the 2018 references.
5. Use vector graphics formats such as PDF instead of raster formats like PNG.

**Questions:**

Questions:
1. In lines 017-018, the abstract emphasizes “however, research on incorporating LLMs for automatic document generation remains limited.” This statement is clearly not accurate.
2. I do not understand the relevance of mentioning hallucination in lines 047-050, as it is not an issue addressed by this paper.
3. Why is PEFT discussed in the related work section? It is not closely related to the main content of the paper. The focus should be on related work concerning automatic literature review generation.
4. Line 121: What is T_u?
5. Line 128: The term PLM should be defined before using the abbreviation.
6. Equation (1): What are p_theta and r?
7. Equation (2): Does R increase as the number of neighboring nodes of v increases?
8. Line 232: What does G_l^’ belongs G_l mean?
9. Are there examples of generated surveys? How do they compare to the standard surveys?
10. Are there examples of the generated hierarchical structure?

---

### Official Review · Reviewer_2de8 · 2024-11-02

**Soundness:** 2
**Presentation:** 3
**Contribution:** 2
**Rating:** 5
**Confidence:** 4

**Summary:**

The paper presents HiReview, an automated framework for literature review generation that leverages a taxonomy-based structure. HiReview first employs retrieval techniques to gather relevant papers from a citation network, followed by hierarchical clustering to construct a topic taxonomy. This taxonomy is refined through joint training of the clustering and topic generation models, ensuring that clusters and topics align meaningfully. Finally, the structured taxonomy is used as input to a large language model (LLM), which generates the literature review content.

The main contributions of this paper are as follows:

1. Introduction of a taxonomy-based literature review generator that organizes topics into a meaningful hierarchy.

2. Joint training of the literature generator and hierarchical clustering model to enhance consistency between clusters and topics.

3. Demonstration that HiReview outperforms existing methods, including LLMs and AutoSurvey, in generating coherent and comprehensive literature reviews.

**Strengths:**

Strengths
1. The framework for generating scientific literature reviews is highly valuable for researchers, especially given the rapid increase in the volume of related papers on any given topic. Organizing the literature review in a taxonomy structure enhances clarity, allowing researchers to navigate the topics more intuitively.

2. The joint training of a graph neural network (GNN) for clustering and a pretrained language model (PLM) for topic generation is a novel and effective approach. This integration ensures that the clusters and topics are consistent and meaningful, which significantly enhances the utility of the generated taxonomy.

3. The framework is presented in a clear and structured manner, making it easy for readers to understand the methodology and follow the proposed workflow.

4. Experimental results demonstrate substantial improvements over baseline methods, validating the effectiveness of the HiReview framework in both structure and content generation quality.

**Weaknesses:**

1. There are already multiple existing topic taxonomies, such as the MAG labels taxonomy. The paper would benefit from a clearer explanation of the necessity for constructing a customized taxonomy within HiReview. Justifying this decision would clarify the unique contribution of the framework in this area.

2. Topic taxonomy construction is a well-researched field with numerous established baselines. However, the paper does not include a comparative analysis against these baselines. Adding such comparisons would provide a stronger foundation for evaluating HiReview’s performance and demonstrate its advantages or limitations relative to existing methods.

3. The paper lacks a comparison of the inference cost between HiReview and previous baseline methods. Given the potential computational demands of joint training and large-scale data processing, it would be beneficial to assess the efficiency of HiReview against alternative frameworks.

4. There is insufficient detail regarding the granularity and structure of the input topic query Q. Clarifying this aspect would help in understanding the adaptability and scalability of HiReview in handling diverse or complex topics.

**Questions:**

1. What is the quality of the generated taxonomy in HiReview compare to existing manually created (CS) label taxonomies?

2. What is the generated taxonomy quality compare to that of existing topic taxonomy construction frameworks?

3. What are the training and inference costs of HiReview in comparison to baseline methods?

4. The reviewer attempted to access the provided code and datasets but was unable to access the repository. Could the authors provide a working link or repository access for reproducibility?

5. Could you provide examples of input queries used in HiReview? Are these queries reflective of real-world application scenarios? (The review has attempted to access the code and data using the given link, but has failed)

6. Is HiReview robust to noise in the input query?

---

### Official Review · Reviewer_Hbkj · 2024-11-02

**Soundness:** 3
**Presentation:** 2
**Contribution:** 3
**Rating:** 3
**Confidence:** 4

**Summary:**

This paper presents HiReview, a framework designed for automatic literature review generation driven by hierarchical taxonomy. The approach begins by retrieving relevant academic documents and organizing them into a structured taxonomy using graph-based hierarchical clustering that considers both textual content and citation relationships. This structure aids in understanding the thematic connections between research papers. The second stage employs retrieval-augmented generation, where a large language model is used to generate comprehensive and citation-aware summaries for each identified cluster. The proposed framework aims to enhance literature review processes by maintaining coherence, ensuring content relevance, and supporting structured synthesis of scientific fields.

**Strengths:**

The paper addresses some important needs deriving from the ever increasing number of publications and the load on researchers to produce accurate and complete states of the art.
The use of a graph-based hierarchical clustering algorithm to structure literature into thematic categories is a relatively novel approach, differentiating this work from existing review generation methods.
The proposed evaluation framework is thorough, assessing the various sub-components of the model. This provides insights into the individual contributions of each part of the system to the overall performance.

**Weaknesses:**

The whole pipeline has a high reliance on LLMs, both for the construction of the content and the evaluation itself. The results may therefore have high variance depending on the model used in the different stages.
In general the pipeline is relatively complex which may pose a problem for reproducibility of the results.
However, one of the main issues in my opinion is that the evaluation does not take into account variations in the research domains. Different research domains may have different ways to express state of the art and cite differently related works, the paper does not contain enough information to assess whether this approach would be effective across different domains.
Another important issue that is also deriving from the extensive use of LLMs, is that there is no evaluation of hallucination risks, or the evaluation of incorrect information. For instance, in the appendix we can see that on one hand the human classification puts "continual TC" under "continual fine-tuning" as the hierarchy is specific to LLMs, but the automatically built taxonomy puts "continual TC" under "applications of continual learning" as if the hierarchy was a general one and not specific to LLMs. This example also highlights how difficult is the evaluation of these hierarchies and probably measures such as bertScore or LLMScore are hiding the details, which may be important in an application like this.
Also, the sentence "Distillation-based methods focus on transferring knowledge from previous models to new ones" in the example in the appendix is not completely correct because distillation is more about the knowledge transfer from teachers to learners.

**Questions:**

Some minor corrections:

Figure 1: Retreive -> Retrieve
Table 3: HiCluserting -> HiClustering
also Table 3: Accuracy evaluated on what?
Table 2: why only LLMScores are reported?

text encoder to convert titles : which one?

RQs seem more engineering questions that research ones (and RQ3 has no explicit answer)

Section 5.1 are the articles in the same domain or do they cover different domains?

---

### Official Review · Reviewer_RiwN · 2024-11-04

**Soundness:** 3
**Presentation:** 3
**Contribution:** 3
**Rating:** 6
**Confidence:** 3

**Summary:**

This paper works on the literature review generation task. Given a set of papers under a shared research topic, the task is to generate a structured summary of them similar to a human written literature survey paper. The major contribution of the paper is a graph learning based hierarchical clustering approach, which utilizes the citation graph and textual information (paper titles) to iteratively form paper clusters to reflect their topics at different levels. When generating the literature survey, the generation process is guided by the taxonomy structure and a set of representative papers. Experiment results show the proposed method outperforms naive LLM prompting, RAG, and existing methods.

**Strengths:**

- A learning-based scientific paper hierarchical clustering method that takes into account both textual evidences and network structure. Could be interesting as an individual topic for further exploration.
- The taxonomy-guided literature review method is also interesting, which large mimic how human write literature review by first organizing the concepts in a hierarchical structure.
- The paper is clearly written

**Weaknesses:**

- The experiment results only show the performance judged by LLMs. However, relying on LLMs to provide a score (out of 100 as shown in the prompts) may not always be reliable, specifically with such kind of long outputs. For example, is it guaranteed a 80-scored passage is always better than a 75-scored passage.
- Some short examples showing how the actual final outputs look like by different methods could be helpful.
- The experiment setting of this paper and the baseline is not strictly fair, in the sense that AutoSurvey retrieves papers from a database while this paper use the 2-hop neighbor network of the human generated survey. The earlier one is a more realistic setting to me, because writing a literature review needs to find relevant papers.
- The paper needs some proofreads as multiple typos exist. For example, at the end of page 8, 'opf' -> 'of'

**Questions:**

See weaknesses. I mainly have two questions:
- How the LLM as scorer can be guaranteed in the experiments? Will it be better to directly compare proposed method with the strongest baseline side-by-side?
- Is there any justification for the current setting? What if there is no 2-hop neighbor network available and the model only has access to some automatically retrieved papers?

---

### Meta-Review · Area_Chair_nGqs · 2024-12-20

**Metareview:**

The paper proposes HiReview - an automated literature review generation, aiming to produce structured and coherent summaries of research papers under a shared topic. HiReview organizes research papers into a hierarchical taxonomy using a graph-based clustering method that incorporates citation relationships and textual content. The taxonomy guides the generation of the literature review content. Papers are clustered iteratively based on their thematic connections, leveraging citation graphs and textual information. The hierarchical clustering and topic generation models are trained jointly to ensure meaningful alignment between clusters and topics. An LLM is used to generate comprehensive, citation-aware summaries for each identified cluster. HiReview is validated on a constructed dataset, demonstrating its effectiveness in generating comprehensive and coherent literature reviews.

**Strengths identified**:
1. The paper tackles the growing challenge of managing and synthesizing the increasing volume of scientific publications, providing researchers with an automated and structured literature review framework.

2. The use of a taxonomy-guided approach that mimics how humans write literature reviews by organizing concepts hierarchically is both novel and intuitive. Graph-based hierarchical clustering incorporating textual and citation network data differentiates this work from existing methods.

3. The proposed taxonomy-then-generation approach demonstrates substantial improvements over baseline methods in generating structured and coherent literature reviews.

**Identified weaknesses that need to be addressed**:
1. The framework heavily relies on LLMs for both content generation and evaluation. This introduces potential bias and variability in results depending on the LLM used at different stages.

2. Lack of human evaluation to validate the outputs, relying instead on automated metrics (LLMScore, BERTScore), which may hide critical details or fail to capture nuances like hallucinations or factual inaccuracies.

3. Insufficient Baseline Comparisons: Missing comparisons with existing topic taxonomy systems, such as MAG labels taxonomy, and other hierarchical literature review frameworks. Clustering methods are not compared with alternatives like GraphRAG, limiting insights into the effectiveness of HiReview’s clustering approach.

4. Risks of hallucination and inaccuracies in generated hierarchies or content are not evaluated. Misclassifications, such as “continual TC” under inconsistent categories, highlight potential issues with automatic taxonomy construction.

**Additional Comments On Reviewer Discussion:**

Authors did not engage during the rebuttal phase.

---

### Decision · Program_Chairs · 2025-01-22

Reject